# Effect of Mulberry Leaf TMR on Growth Performance, Meat Quality and Expression of Meat Quality Master Genes (*ADSL*, *H-FABP*) in Crossbred Black Goats

**DOI:** 10.3390/foods11244032

**Published:** 2022-12-14

**Authors:** Yong Long, Yong Han, Yuanfeng Zhao, Dianqian Chen, Defeng Wang, Yang Yang, Chaozhi Su, Xiaoyun Shen

**Affiliations:** 1Key Laboratory of Animal Genetics, Breeding and Reproduction in the Plateau Mountainous Region, Ministry of Education, Guizhou University, Guiyang 550025, China; 2Institute of Animal Husbandry and Veterinary Sciences, Guizhou Academy of Agricultural Sciences, Guiyang 550025, China

**Keywords:** mulberry leaf, TMR, goat meat, fatty acids, amino acid, meat quality

## Abstract

This study was conducted to examine the effect of a mulberry leaf total mixed ration (TMR) diet on growth performance, apparent digestibility, meat quality and the expression of related meat-quality genes (*ADSL, H-FABP*) in crossbred black goats. Forty-four Guizhou crossbred black goats (Nubian black goat ♂ × Guizhou black goat ♀), weighing 33.43 ± 0.55 kg, were chosen. The goats were randomly divided into four groups, with 11 test replicates in each group. Group I was the control group and fed with the traditional feeding method of roughage and concentrate supplement without adding mulberry leaf. Group II was fed with a 40% mulberry leaf pellet TMR diet. Group III was fed with a freshly processed 40% mulberry leaf TMR diet. Group IV was fed with a 40% mulberry leaf fermented total mixed rations (FTMR) diet. The results showed that the average daily gain (ADG) of group II was significantly higher than that of group I and III (*p* < 0.05). The apparent digestibility of group II of ether extract (EE) and neutral detergent fiber (NDF) was significantly higher than that of group I (*p* < 0.05), and the apparent digestibility of dry matter (DM) and crude protein (CP) was significantly higher than that of group I (*p* < 0.01). Compared with group I, meat in group II had lower meat color lightness (L*) and yellowness (b*) values (*p* < 0.01) in the Longissimus thoracis et lumborum. The shear force of group II was significantly lower than that of group I (*p* < 0.05). The total fatty acids (TFA) of group II was significantly higher than that of groups I and III (*p* < 0.05), but the total saturated fatty acids (SFA) of group II was significantly lower that than of group I (*p* < 0.01). Subsequently, the Unsaturated fatty acids (USFA), Monounsaturated fatty acids (MUFA), and Polyunsaturated fatty acids (PUFA) of group II were significantly higher than those in group I (*p* < 0.01). The contents of total amino acids (TAA), total essential amino acids (EAA), total non-essential amino acids (NEAA) and total of major fresh-tasting amino acids (DAA) of groups II, III and IV were significantly higher than those of group I (*p* < 0.05), as well as the contents of IMP (*p* < 0.01). The expression of the *H-FABP* gene in the arm triceps of group II was significantly higher than that of groups I, III and IV (*p* < 0.05). The expression of the *ADSL* gene in the Longissimus thoracis et lumborum and biceps femoris of group II was significantly higher than that of group I (*p* < 0.05). Collectively, the results of the current study indicated that the mulberry leaf TMR diet improved the growth performance, apparent digestibility and expression of related meat-quality master genes (*ADSL, H-FABP*) in crossbred black goats, which promoted the deposition of intramuscular fat (IMF) and inosinic acid (IMP) and improved the composition of fatty acids and amino acids in the muscles.

## 1. Introduction

With a wide diversity of amino acids and a low level of cholesterol, goat meat has been increasingly popular due to its distinct flavor and tenderness. The tenderness and taste of goat meat depend on the amount of fat accumulated within the muscle tissue [1]. However, relevant studies show that intramuscular fat (IMF) can be directly involved in the formation of muscle tenderness and juiciness and is one of the important indicators of meat tenderness and flavor [2,3]. A higher IMF content can increase consumers’ satisfaction with the quality of the goat meat [4]. The IMF content and the heart fatty-acid-binding protein (*H-FABP*) gene expression levels were shown to be positively correlated [5]. Some studies have shown that there is a significant correlation between the umami of goat meat and the amino acid composition and inosinic acid (inosine monophosphate, IMP) content in the muscle [6,7]. Adenylosuccinate lyase (*ADSL*) is one of the major enzymes deposited by IMP in animal organisms and is encoded by the *ADSL* gene [8]. However, studies on the effect of the *ADSL* gene on IMP are mostly reported in poultry, and studies in ruminants are relatively rare. The binding of IMP to Asp and Glu enhances goat meat umami [9], consequently improving the taste of the meat, which is one of the important indicators of the umami assessment of meat [10,11]. It has been shown that the expression level of *H-FABP* and *ADSL* genes can affect the content of IMF and IMP in meat, and indirectly participate in the formation of muscle tenderness, juiciness, flavor and freshness [8,12,13].

*Mulberry alba* belongs to the Moraceae family (Morus) and is deciduous woody perennial that is widely planted because of its strong resistance to adversity and other advantages [14]. Mulberry leaves are rich in nutrients and have a large number of unique bioactive substances, mainly including flavonoids, steroids and alkaloids, which can be used as high-quality livestock feed [15,16]. The addition of mulberry leaf to animal diets can improve their growth performance, anti-inflammatory activity, and immunity. Furthermore, it can regulate the rumen fermentation of ruminants, promote the digestion and absorption of nutrients and improve meat quality [17,18,19]. However, the mulberry leaf has high crude fiber content; in addition, the content of phytic acid and tannin can be as high as 488.9 mg/kg and 5.32 mg/kg, respectively. The high content of anti-nutritional factors can reduce the digestion and absorption of nutrients, so the addition of an excessive amount of anti-nutritional factors in the diet will affect the animal’s production performance and bodily health [20,21,22]. 

Studies on the mulberry leaf as a ruminant feed have been reported for a long time, but the literature on the effects of total mixed rations (TMR) diets with mulberry leaf on the growth performance and meat quality of black goats are extremely rare. Previously, our group has conducted a field single-cage feeding test, by adding different proportions (30%, 40%, 50%) of mulberry leaf to a TMR diet to feed black goats, and found that the addition of 40% can significantly reduce the serum total cholesterol (TC) content and increase the serum T3 and T4 content in black goats. In addition, the 40% mulberry leaf TMR group significantly reduced the NH3-N content in the rumen fluid of black goats and significantly increased the feed intake, total weight gain, daily gain and economic benefit of black goats, as well as reduced the feed-to-meat ratio. The black goats fed with the 40% mulberry leaf TMR diet had the best growth performance, economic benefit, and fattening effect among all the groups [23]. Mulberry leaves are rich in aldehydes, alcohols, esters and ketones, which are the main characteristic compounds of meat flavor, and feeding goats mulberry leaf can improve the composition of fatty acids and amino acids in the meat, which can enhance the flavor of the meat [24]. To further investigate the effects of 40% mulberry leaf as roughage in the diets of black goats, a comparative analysis of feed nutrition was conducted in this study. Moreover, the effects of the different processing methods of the mulberry leaf TMR diet on the growth performance, apparent digestibility, and meat quality of black goats were characterized comprehensively.

## 2. Materials and Methods

This experiment was carried out at the Guizhou Jels Recycling Agriculture Co. (Sandu, Guizhou, China) with the following geographical characteristics: a latitude of 107°87′54″ S, a longitude of 25°98′15″ W, an altitude of 399 m, an average annual rainfall of 1350 mm, and an average temperature 17.5 °C. All animal procedures were performed with the strictest adherence to animal welfare guidelines and with regulatory oversight by the Experimental Animal Ethics Committee of Guizhou University in Guizhou, China (EAE-GZU-2020-P028).

### 2.1. Materials, Animals, Diets and Experimental Design

The TMR formulation was redesigned based on the optimum mulberry leaf level from a previous study [23]. Forty-four healthy, 5–6 months old, crossbred growing rams (Nubian ♂ × Guizhou black goat ♀) with 33.43 ± 0.55 kg live body weight were used in the experiment. The goats were divided into four groups by using the RCBD experimental design, each group with 11 test replicates. Experimental group I was the control group and fed with the traditional feeding method of roughage and concentrate supplement without adding mulberry leaf. The feed for test group II was mixed with all ingredients after adding 40% mulberry leaf according to the designed formula. A small pelletizer (DKS-150B, Henan Tongda Heavy Industry Technology Co. China, Zhengzhou, China) was used to make TMR into pellet TMR. The feed for group III was prepared by adding 40% mulberry leaf in the designed formula and then mixing the feed ingredients evenly to make TMR. The feed for group IV was made into fermented total mixed rations (FTMRs) by adding 40% mulberry leaf according to the designed formula and then fermenting the TMRs, which were wrapped and sealed for 15 d using a fully automatic silage round baler (MK5050-G, Leling Guoyu Agricultural and Animal Husbandry Machinery and Equipment Co., Ltd., Qingzhou, China) with a moisture content of 42% and a compression ratio of 480 kg/m^3^. The experimental period lasted 85 days, with 10 days of adaptation and 75 days of data and sample collection. The base diet was formulated according to the NRC’s (2012) nutritional requirements, and its composition and nutrient levels are shown in Table 1.

The experimental goats that were housed in individual stalls (2 m^2^) were disinfected, dewormed and immunized according to the routine procedures of the black goat farm. The diets were the same for each test group in the pre-test and formal test periods. Feed was offered ad libitum to the experiment goats at 8:00 and 16:00. The feed was adjusted daily according to the remaining feed from the previous day to ensure that about 1% of the feed remained in the feed trough at the second feeding.

### 2.2. Sample Collection and Apparent Digestibility

Feces were collected daily from day 70 to day 75, twice a day, with 20 g of feces each time. After 5 days of continuous collection, the manure samples were mixed in equal amounts with a ratio of 20% (*w*/*w*) of fresh manure weight, 10% dilute sulfuric acid was added to fix nitrogen and it was stored in a −20 °C refrigerator.

Samples of feed and feces were dried in a forced-air oven at 65 °C for 48 h and crushed samples were grounded to pass through a 1 mm screen stored in hermetic bag until analysis [25]. Dried samples were used for the detection of dry matter (DM, method No. 934.01), crude protein (CP, method No. 954.01), and ether extract (EE, method No. 920.39) according to AOAC [26]. The neutral detergent fiber (NDF) of the feed was determined according to Van Soest et al. [27]. The nutrient digestibility was calculated by using the endogenous indicator acid insoluble ash (AIA) with the same determination criteria, using the following formula:Apparent digestibility of a nutrient = [1 − (Indicator content in feed/Indicator content in manure) × (Content of a nutrient in manure/Content of that nutrient in feed)] × 100%

### 2.3. Growth Performance and Economic Benefits 

The average daily gain (ADG) was calculated every 15 days on a regular basis. The troughs were cleaned and weighed before feeding each day. Feed and residuals were collected and the dry matter intake (DMI) was calculated. A feed weight ratio (FWR) was calculated based on the DMI and ADG. ADG = (end of test weight − beginning of test weight/number of feeding days); FWR = DMI/ADG; DMI (kg/d) = [feeding amount (kg) × diet DM (%) − leftover (kg) × leftover DM (%)]/[number of black goat per group × number of trial days (d)]; Feed weight gain cost (CNY/kg) = total feed intake × feed unit price (CNY)/total weight gain; Weight gain benefit (CNY/one goat) = (real-time price of live black goat − feed weight gain cost) × total weight gain.

### 2.4. Meat Quality Profiles and Nutritional Composition Profiles

At the end of the experiment, 5 black goats were randomly selected from each treatment (20 in total) and were fasted for 12 h before being slaughtered to determine slaughter traits and meat quality. According to animal welfare requirements as per Prescott and Lidster [28], black goats were knocked out and hung up. The jugular vein and carotid artery were cut, and the organs, forelimb knee joints and hind limb toe joints were removed. The dressed carcasses were weighed within 1 h (hot carcass weight) [29]. 

Samples of Longissimus thoracis et lumborum, biceps femoris and arm triceps were collected (about 0.5 kg of each), vacuum-packed, snap-frozen in liquid nitrogen and stored in the freezer at −80 °C. The nutritional composition, amino acid types and contents, fatty acid types and contents, and inosinic acid (inosine monophosphate, IMP) content of the meats were analyzed. Longissimus thoracis et lumborum was cut from the mid-section and the surface fascia was removed. A pH meter (pHS-3E, Shanghai Yidian Scientific Instruments Co., Ltd., Shanghai, China) was used to measure the pH value of Longissimus thoracis et lumborum 45 min and 24 h after slaughter. The meat color of lamb was measured with a colorimeter (model Minolta CR400, Minolta Inc., Osaka, Japan). The brightness, redness and yellowness of meats were indicated by L*, a*, and b*, respectively, and the final average was taken after three measurements for each sample [30]. Samples were steamed in a preheated water bath to measure the cooking yield according to the method of Abhijith et al. [31]. Shear force was determined using the same samples used earlier for the cooking loss measurement and according to the method of Garba et al. [32]. Water loss rate was determined according to the method of Parente et al. [30]. Meat samples with an area of about 5 cm^2^ were wrapped in multiple layers of filter paper and weighed after being pressed with a 35 kg iron block for 5 min. The percentage of the difference between the pre-press and post-press weight was the water loss rate. When the core temperature reached 70 °C, the samples were taken out and cooled in cold water to prevent further cooking, and then patted dry with lint-free paper towels.

Regarding the chemical composition of meat, the CP (Method No. 981.10), moisture content (Method No. 925.04), ash (Method No. 938.08) and ether extract (Method No. 935.38) were determined according to methodologies described by the AOAC [33].

### 2.5. Fatty Acid Profiles

The same masses of Longissimus thoracis et lumborum, biceps femoris, and arm triceps were mixed, grounded and then dried. A total of 1.70 g of the mixed goat meat dry sample was analyzed qualitatively followed by a quantitative analysis of 37 fatty acids by gas chromatograph (GCMS-SQ8T, PE companies).

Chromatographic conditions were as follows: Capillary column SH-Rt-2560 (100 m × 0.25 mm × 0.20 µm), purchased from Japan Shimadzu Enterprise Management Co., Ltd., in constant flow mode. The heating up procedure was as follows: the starting temperature was 100 °C held for 13 min, increased at 10 °C/min to 180 °C and held for 8 min, increased at 1 °C/min to 200 °C and held for 20 min, and then increased at 4 °C/min to 230 °C and held for 12 min. High Helium was used as the carrier gas at a flow rate of 1 mL/min. The MS conditions were as follows: The electron bombardment ion source (EI) technique was used, with 70 eV for the emission voltage, 200 °C for the ion source temperature, 220 °C for the transmission line temperature, and a scan range of 35 ms to 450 ms.

### 2.6. Amino Acid Profiles 

The same masses of Longissimus thoracis et lumborum, biceps femoris and arm triceps were mixed and grounded, and then dried. About 100 mg of the sample was put into a 20 mL glass hydrolysis tube. A solution of 10 mL of 6.0 mol/L hydrochloric acid was added, shaken well and pumped to a vacuum. The glass hydrolysis tubes were sealed and placed in a constant-temperature oven and hydrolyzed at 110 ± 2 °C for 22 h. After cooling, the pH was adjusted to be neutral with 6.0 mol/L sodium hydroxide followed by adding distilled water to 25 mL. Furthermore, 1 mL of sample solution was mixed with 1 mL of 0.02 mol/L hydrochloric acid solution and filtered through a 0.45 μm filter. The content of each amino acid was analyzed by an amino acid analyzer (L-8900, Hitachi, Tokyo, Japan).

Chromatographic conditions were as follows: standard analytical column (4.6 mm × 60.0 mm), separation column temperature: 57.0 °C, and reaction column temperature: 136 °C.

### 2.7. Inosinic Acid Profiles 

A total of 20 mL of 5% perchloric acid was added to 2 g of crushed mixed meat samples of Longissimus thoracis et lumborum, biceps femoris, and arm triceps for the ultrasonic treatment for 20 min. Furthermore, samples were transferred to a 50 mL volumetric flask. A total of 20 mL of the sample solution was placed in a centrifuge tube and centrifuged at 2400 r/min for 4 min. Then 10 mL of supernatant was taken. The pH was adjusted to 6.5 with 0.5 mol/L sodium hydroxide (4 g of sodium hydroxide dissolved in 200 mL of water). We added distilled water to 50 mL in a volumetric flask, shook it well and then analyzed it on the machine.

Chromatographic column had the following specifications: Topsil C18 Colum, 4.6 mm × 250 mm, 5 μm, Lot Number: T2101.10. The detection wavelength of the external detector was 254 nm, and the injection volume was 10 μL. Mobile phase specifications were as follows: flow rate was 1 mL/min, 0.05 mol/L, ammonium formate solution (adjusted pH to 6.5): methanol = 95:5, the detection time was 6 min and the analytical column temperature was 30 °C.

### 2.8. Gene Expression

Extraction of total RNA from the Longissimus thoracis et lumborum, biceps femoris and arm triceps of black goats was performed by the Trizol method. To confirm the quality and quantity of the extracted RNA, samples were analyzed by a NanoDrop One 2000 spectrophotometer (Thermo Scientific, Waltham, MA, USA). Then, cDNA was generated by utilizing the cDNA Reverse Transcription Kit (Beijing Kangwei Century Biotechnology Co., Beijing, China). Reverse transcription was carried out in a 20 μL reaction with 1 μL of HiFiScript, 200 U/μL, 1 μLof Primer Mix, and 4 μL of 5×ScriptRT Buffer. RNase-free water was added to obtain 20 μL. The reverse transcription conditions were set as follows: incubation at 42 °C held for 15 min, and incubation at 85 °C held for 5 min. Then, the obtained cDNA was stored at −20 °C for later use.

Glyceraldehyde-3-phosphate dehydrogenase (GADPH) was used as the internal reference gene. According to the *H-FABP* gene sequence (accession number: NM_001285701.1) and *ADSL* gene sequence (accession number: XM_005681096.2) on GenBank, the primers for the gene expression analysis were newly designed with the aid of the Primer Premier 5, and are shown in Table 2.

Quantitative PCR was carried out in a 10 μL reaction volume, containing 0.5 μL of each forward and reverse primer, 5 μL of 2×UltraSYBR Mixture (Beijing Kangwei Century Biotechnology Co.), 0.5 μL of cDNA (500 ng/μL) template, and addition of ddH_2_O to 10 μL. RT-PCR amplification procedure consisted of an initial denaturation step at 95 °C for 10 min followed by 45 cycles of 95 °C for 15 s and 55 °C for 1 min for *ADSL*, *H-FABP* and *GAPDH*. Quantitative PCR reactions were performed in triplicate for each sample type/primer set combination. The level of expression of each gene relative to that of *GAPDH* was evaluated by the 2^−ΔΔCt^ method described by Livak and Schmittgen [34].

### 2.9. Statistical Analysis

Data were compiled using Excel 2019 and statistically analyzed using SPSS l9.0 software. The Shapiro-Wilk test was firstly applied to determine the normal distribution of the data. This was followed by one-way ANOVA and general linear model (GLM module) for multi-covariate ANOVA. Multiple comparisons and significant difference tests were performed by using the LSD method and Duncan’s test. The table in the text shows the mean and standard error (SEM) of each group. *p* < 0.05 was considered significant and *p* < 0.01 was considered extremely significant.

## 3. Results

### 3.1. Growth Performance and Apparent Digestibility

The DMI of group II was significantly higher than that of the other groups (*p* < 0.01). Meanwhile, the ADG of group II was significantly higher than groups I and III (*p* < 0.05), and the difference with group IV was not significantly influenced (*p* > 0.05). Compared with group I, the FWR in group II was reduced by 9.53% (*p* > 0.05). The total weight gain of group II was significantly higher than that in groups I and III (*p* < 0.05). The feed weight gain cost of group II was significantly higher than that in group III (*p* < 0.05). The weight gain benefit among the groups was in the following order: group II > group III > group IV > group I (Table 3).

Compared with the control group, the apparent digestibility of DM in groups II and III was significantly increased (*p* < 0.01), and the apparent digestibility of EE in group II increased by 4.32% (*p* < 0.05). There was no significant difference between groups II, III and IV (*p* > 0.05). The apparent digestibility of crude protein in group II was significantly higher than that in group I (*p* < 0.01). In addition, the apparent digestibility of neutral detergent fiber (NDF) in groups II and III was significantly higher than that in group I (*p* < 0.05) (Table 4).

### 3.2. Meat Quality

The Longissimus thoracis et lumborum goat meat color L* value of group II was significantly lower than that of the other groups (*p* < 0.01), and the b* value was significantly lower than that of group I (*p* < 0.01). However, the difference in the a* value between the test groups was not significant (*p* > 0.05). In arm triceps, the L* value of group II was significantly lower than that in group I (*p* < 0.05). Meanwhile, the b* value was significantly lower than that in group I (*p* < 0.01), and the a* value between test groups was not significantly different (*p* > 0.05). In the biceps femoris, no difference was observed in the L* value and b* value between the test groups (*p* > 0.05). However, the a* values in group II were significantly higher than those in group I (*p* < 0.05). The pH of the goat meat of group I at 45 min was higher than the other groups (*p* < 0.05), and there was no difference in the pH at 24 h among the groups (*p* > 0.05). Group II with the highest value was more stable in terms of pH. In the Longissimus thoracis et lumborum and the arm triceps, the sheer force of group II was significantly lower than that in group I (*p* < 0.05), while no difference was observed in the rest of the experimental groups (*p* > 0.05) (Table 5).

### 3.3. Goat Meat Nutritionl Composition Profiles

The MC and EE of groups II and IV were significantly higher than those in group I (*p* < 0.01). However, the CP was significantly higher in group II than that in groups I and III (*p* < 0.01). Additionally, no difference was observed in the MC, EE and ash between groups II, III and IV (*p* > 0.05) (Table 6).

### 3.4. Fatty Acid Profiles

As shown in Table 7, a total of 25 fatty acids were detected in this test, including 12 saturated fatty acids and 13 unsaturated fatty acids; however, some fatty acids were not detected due to low content. The C18:1n9n, C18:2n6 and C20:3n6 in group I were significantly lower than those in other groups (*p* < 0.01), whereas C20:4n6 was significantly lower than that in other groups (*p* < 0.05). Additionally, the C8:0, C11:0 and C14:1 of group I were significantly higher than the other groups (*p* < 0.01). Meanwhile, the C15:0 was significantly higher than that in groups II and III (*p* < 0.01), but not group IV (*p* > 0.05). The C16:0 and C18:0 of groups I and III were significantly higher than those in groups II and IV (*p* < 0.01). The C16:1, C17:1 and C18:1n9n of group II were significantly higher than those of the other groups (*p* < 0.01). However, the C4:0 was significantly lower than that in other groups (*p* < 0.01). There was no significant difference in the remaining detected fatty acids when comparing groups II, III and IV with group I (*p* > 0.05) (Table 7).

Compared to group I, the TFA of group II was significantly increased by 10.33% (*p* < 0.05), and the SFA of groups II, III and IV were significantly reduced by 22.35%, 15.82% and 11.90% (*p* < 0.01), respectively. Moreover, the USFA in groups II, III and IV were increased, respectively, by 50.90%, 19.81% and 25.19% (*p* < 0.01). However, when compared with group I, the MUFA increased by 56.47% and 22.28% in groups II and IV, respectively, and the PUFA increased by 36.71% and 39.58% in groups II and IV, respectively (Table 7).

### 3.5. Amino Acid Profiles

A total of 17 kinds of amino acids were determined in this experiment. From the table, it can be seen that the contents of Cys and Met in black goats in each group were not significantly different among the groups (*p* > 0.05). The contents of Asp, Thr, Glu, Gly, Ala, Iie, Leu, Tyr, Phe, Lys, His, Arg and Pro of black goat meats in groups II and IV were significantly higher than those in group I (*p* < 0.05), but not significantly different from those in group III (*p* > 0.05). The contents of Ser and Val were significantly higher in groups II, III and IV than those in group I (*p* < 0.05), while the differences between the remaining groups were not significant (*p* > 0.05) (Table 8).

Compared with group I, the addition of mulberry leaf to the diet of black goats significantly increased the content of TAA, EAA, NEAA and DAA in the goat meat. Particularly, group II showed the most improvement with increases of 19.87%, 21.94%, 21.04% and 23.11%, respectively (*p* < 0.05) (Table 8).

The contents of TAA, EAA and DAA in group II were the highest. Compared with group I, TAA, EAA, NEAA and DAA in group II were significantly increased by 20.96%, 21.90%, 20.55% and 20.41%, respectively (*p* < 0.05). Compared with groups III and IV, the TAA of group II increased by 8.41% and 0.74%, respectively, and the EAA increased by 9.48% and 0.13%, respectively. The DAA content in group II was significantly increased by 17.68% compared to that of the control group (*p* < 0.05). Compared with group III, the content of TAA in group IV decreased by 7.6%, while the contents of EAA, NEAA and DAA increased by 8.04%, 9.66%, and 9.40%, respectively. Compared with group I, the contents of TAA, EAA, NEAA and DAA in group III increased by 11.59%, 11.34%, 11.7% and 12.21%, respectively, but the differences were not significant (*p* > 0.05). Compared with group I, the contents of TAA, EAA, NEAA and DAA in group IV were significantly increased by 20.07%, 20.30%, 22.57% and 22.76%, respectively (*p* < 0.05) (Table 8).

### 3.6. Inosinic Acid Analysis

The inosinic acid contents of groups II, III and IV were significantly higher than those of the control group (*p* < 0.01). Meanwhile, the inosinic acid content of group III was significantly lower than that of groups II and IV (*p* < 0.01). The difference in inosinic acid content between group II and III was not significant (*p* > 0.05) (Figure 1).

### 3.7. Gene Expression

The expression of the *H-FABP* gene in three different muscle tissues of black goats fed with different processing methods of the mulberry leaf TMR diet varied. In the Longissimus thoracis et lumborum, group II had the highest expression of the *H-FABP* gene, which was significantly higher than that of group III (*p* < 0.05), but not groups I and IV (*p* > 0.05). The expression of the *H-FABP* gene in the arm triceps muscle of group II was significantly higher than that of the other groups (*p* < 0.05). However, the expression of the *H-FABP* gene in the biceps femoris was not significant among all groups (*p* > 0.05) (Figure 2).

In the Longissimus thoracis et lumborum and biceps femoris, the *ADSL* gene expression of group II was significantly higher than that of group I (*p* < 0.05), but no difference was observed between groups III and IV (*p* > 0.05). In the biceps femoris, there was no significant difference in *ADSL* gene expression among the groups (*p* > 0.05) (Figure 2).

## 4. Discussion

An animal’s apparent digestibility, productive performance and carcass traits are influenced by breed, type, structure and feed processing method. Studies have shown that an increase in feed intake was related to a feed containing special substances that increased the number and efficiency of rumen microorganisms and improved the digestibility of the diet, eventually leading to an increase in feed intake [35]. Liu et al. [36] found that the addition of mulberry leaf to lamb diets could achieve the purpose of improving the rumen microbial environment and increasing feed intake. In the present study, the addition of mulberry leaf to the feed improved increased the DMI, ADG and economic efficiency of black goats, which were similar to the findings of a previous study by Jia et al. [37].

The apparent digestibility of nutrients reflects the rationality of the diet formulation and is an important indicator of the balance of nutrients, which is closely related to the growth rate and economic benefits of animals. Huyen et al. [38] found that the digestibility of nutrients, such as DM, CP and EE, increased linearly with an increase in the level of added mulberry leaf pellets in the diet of beef cattle, and similar results were obtained in our study. It is possible that the mulberry leaf TMR diet, through pelleting, increased the intake and overall diet digestibility of black goats, leading to the increased feeding intensity and energy supply. Meanwhile, the pellet TMR diet could promote the absorption and conversion of nutrients, such as protein and fat [17], which would then accelerate muscle tissue production and fat deposition.

Meat color, tenderness, pH and water loss rate are important indicators to identify meat quality and are influenced by breeding, nutrition level, feeding management, etc. Changes in pH are closely associated with biochemical changes in meat. The rate of pH decrease is proportional to the rate of biochemical changes in meat, which plays an important role in assessments of meat quality. The final pH of the animal after slaughter has an important link to meat storage and can directly affect the meat’s shelf life and storage duration [39]. This is due to the accelerated rate of muscle glycolysis and increased glycogen lyase activity in black goats after slaughter, in which ATP is hydrolyzed to produce phosphate. Additionally, a large amount of muscle glycogen is broken down to produce acidic muscle deposits, such as lactic acid, resulting in a decrease in muscle pH and meat spoilage [40,41]. In this study, the reduction rate of pH from 45 min to 24 h after slaughter in black goats fed with added mulberry leaf was slower than that of the control group, which shows that adding mulberry leaf to the diet can slow down the spoilage rate of black goat meat, prolong the shelf life of the meat and improve the meat quality.

Meat color is closely related to myoglobin content and muscle fiber type, which is a major visual determinant of consumers’ consumption decisions [42,43]. Myoglobin can be transformed into oxygenated myoglobin, of which the ratio can directly affect the a* value. Oxygenated myoglobin can be easily oxidized to form high-iron myoglobin during the storage of the meat, which then affects the meat color [44]. The higher the a* value, the better the meat quality; the higher the L* value, the paler the meat color and the worse the meat quality [45]. Chelh et al. [46] found that the oxidation of proteins and lipids produces a Schiff pigment (lipofuscin) that increases the b* value of meat. In this study, the addition of mulberry leaf to the diet increased the a* value and significantly reduced the L* and b* values of the goat meat, in agreement with the findings of Jia et al. [38]. It is possible that the addition of mulberry leaf to the diet could produce a substance that affected the Schiff pigment production during the oxidation of proteins and lipids, which needs to be further studied in depth. In addition, the levels of EE and CP in group II were significantly higher than those of group I, with the highest nutrient content and the most obvious effect in improving the meat quality.

The water-holding capacity of muscle is generally expressed in terms of cooking yield, and the meat with a high cooking yield is juicy and tender. However, drip loss and steaming loss are the main indicators of the water-holding power of muscle. The amount of water-holding power in muscle can influence the juiciness of meat products. Meat tenderness depends on the fat content accumulated within the muscle tissue [1], which can range from 10 to 45% depending on age, sex and diet [47]. Shear force is an important indicator of meat tenderness, and the size of the shear force of meat is inversely proportional to tenderness. In this study, the shear force of the goat meat in groups II, III and IV was lower than that of group I, and the water loss rate and steaming loss were higher than those of group I. Meanwhile, the analysis of goat meat muscle nutrient composition revealed that MC, EE and CP were also higher than those of group I. It may be that mulberry leaves are rich in nutrients and some bioactive components can affect the nutritional components and adipose tissue composition of mutton; further research is needed.

IMF deposition is closely related to the fat content, and a low IMF content can lead to poor tenderness and juiciness of the lamb [48]. The addition of mulberry leaf to the diet can improve nitrogen utilization and facilitate protein deposition, resulting in a higher IMF content [49]. Realini et al. [4] found that IMF deposition improved the meat quality and the fatty acid composition of lambs in a study of consumer preference for lambs from New Zealand pastures. In a study of molecular characteristics and expression patterns of FABP family genes, Wang et al. [5] found that FABP family genes were highly expressed in many tissues, including goat fat, and the IMF content had a significant relationship with the *H-FABP* gene expression levels. In this study, we detected the mRNA expression level of the *H-FABP* gene in the Longissimus thoracis et lumborum, biceps femoris, and arm triceps muscles by RT-PCR and found that the expression level of the *H-FABP* gene was increased in groups II, III and IV. The results suggested that the addition of nutrient-rich mulberry leaf to the diet could improve the feed intake and daily weight gain of black goats, increasing the expression of the *H-FABP* gene in goat meat, changing the IMF deposition and the composition of the fatty acids, and leading to better goat meat qualities with improved tenderness and juiciness.

The level and composition of fatty acids in meat play a decisive role in the meat’s quality and flavor [50], which are mainly influenced by nutrition and feeding time, and to a lesser extent by breed and sex [51,52]. Unsaturated fatty acids are subdivided into monounsaturated fatty acids and polyunsaturated fatty acids, which are the main sources of muscle flavor formation [53]. The heavy off-flavor of goat meat is one of the main factors that affect consumer consumption. The off-flavor is primarily caused by the volatiles produced by the oxidation of saturated fatty acids [54]. Sanudo et al. [55] found that stearic acid (C18:0) and linolenic acid (C18:3) are also the major substances that cause the off-flavor in lamb. Meanwhile, C18:0 is an important saturated fatty acid in goat meat, since changes in the concentration of C18:0 and C16:0 can affect the SFA content in intramuscular fat [52,56]. Consequently, an excessive intake of SFA can increase the levels of high-density lipoprotein (HDL) and low-density lipoprotein (LDL) in human blood, which can lead to the development of coronary atherosclerosis and other diseases [57]. In this study, it was found that feeding TMR diets supplemented with mulberry leaf to black goats reduced the content of short-chain fatty acids, especially C4:0 and C8:0, in the mutton. Moreover, the content of C18:0 and C16:0 was significantly decreased in group II. In addition, feeding goats the mulberry leaf TMR diet significantly reduced the SFA content in the goat meat.

Increased USFA content can improve meat flavor, taste and nutritional value, but too much might cause muscle to oxidize quickly, lowering the meat quality. The saturation of lipids affects the activity of lipid oxidation reactions. PUFA with an attractive flavor has a high nutritional value and can reduce human blood cholesterol levels, which can prevent cardiovascular disease [58]. Oleic acid is a monounsaturated fatty acid (MUFA) that can reduce blood cholesterol and LDL cholesterol levels [59]. Linoleic acid (C18:2n6) is an essential fatty acid that the human body cannot synthesize by itself [60]. Our results in this study showed that diets supplemented with mulberry leaf could increase the content of oleic acid (C18:1n9n) and linoleic acid (C18:2n6). Meanwhile, arachidonic acid (C20:4n6) also had different degrees of improvement. Moreover, the addition of mulberry leaf to the diet of black goats could increase the content of PUFA, MUFA and USFA in lamb, which is consistent with the results of a previous study [61]. It is speculated that certain bioactive substances contained in mulberry leaf have some regulatory effects on lipid metabolism and deposition, which can improve the fatty acid composition and meat quality.

The content and composition of amino acids are the main determinants of meat quality, especially the content of essential amino acid, which contributes positively to the nutritional value of meat and is one of the main sources of the flavor of fresh meat. The Maillard reaction between amino acids and reducing sugars is an important way to generate flavor substances [62]. The composition and content of amino acids in muscle directly affect the nutritional value and flavor of meat [49]. Thr helps to promote growth, maintain body protein equilibrium and regulate the function of endocrine homeostasis [63]. Val can regulate amino acid and lipid metabolism and is related to the nervous and endocrine systems; some studies have shown that adding Val to the diet helps piglets grow [64]. In this study, it was found that feeding goats a mulberry leaf TMR diet significantly increased the Thr and Val levels.

Asp, Glu, Gly, Arg and Tyr are closely related to the meat’s flavor. Glu and Asp are considered the key amino acids that affect the flavor of lamb [65,66]. The content of DAA can directly affect the flavor of lamb [67]. Arg helps to maintain the balance between energy intake and consumption and regulates fatty acid composition, which can suppress obesity and improve meat quality [68]. EAA as an essential amino acid cannot be synthesized by the human body or synthesized in amounts that meet human needs, according to FAO/WHO standards. The EAA/TAA ratio of high-quality proteins is around 40% [48]. In this study, it was concluded that the addition of mulberry leaf to the diet of black goats could increase the content of each amino acid in mutton. Seventeen amino acids were detected in this experiment, with Glu having the highest content, followed by Asp and Lys. The EAA/TAA of each group was between 41 and 43%, indicating that black goat meat is a high-quality protein source. The EAA/TAA ratio of groups II, III and IV was higher than that of group I, suggesting that the protein of the meat of goats fed with the mulberry leaf TMR diet was better than that of goats fed conventional feed. In addition, the levels of TAA, EAA and DAA in group II were significantly higher than those of group I, showing that the mulberry leaf pellet TMR diet is the most effective in improving the content of essential and non-essential amino acids in the rations of black goats.

After slaughtering the black goats, ATP in the muscle decomposes under hypoxic conditions to produce IMP. IMP is a flavor-enhancing substance, which is closely related to the muscle’s umami. In addition, its degradation products combined with certain amino acids can result in a Mailard reaction under heating conditions to produce volatile substances, which can affect meat freshness and are a key factor in determining the umami of meat. Related studies [6,69] have reported that the *ADSL* gene is one of the main enzymes for IMP deposition in animal organisms and plays an important role in IMP synthesis, which is closely related to meat quality [8,70]. A high expression of the *ADSL* gene and the GARS-AIRS-GART gene combination enhances the IMP content in chicken meat [71]. One study showed that the addition of mulberry leaf to the diet of fattening pigs increased the eye muscle area, CP content, and IMP content [49]. In this study, the addition of mulberry leaf to the TMR diet revealed that the IMP content in groups II and IV was significantly higher than that of groups I and III. The expression level of the *ADSL* gene in the Longissimus thoracis et lumborum and arm triceps muscle of group II was significantly higher than that of group I. It is possible that the nutrients and bioactive components rich in mulberry leaf can improve the mRNA expression level of the *ADSL* gene in black goats and increase the IMP content of meat, which can promote the combination of IMP with umami amino acids, such as Asp, Glu and Gly, leading to an increase in the umami of goat meat. In conclusion, the mulberry leaf pellet TMR diet was the most effective in improving the *ADSL* gene expression and the best in improving the flavor of the goat meat.

## 5. Conclusions

In this study, we attempted to unravel the effects of a mulberry leaf TMR diet on the growth performance and meat quality of crossbred black goats. Our findings demon-started that mulberry leaves could improve the apparent digestibility of nutrients in the diet, leading to a better growth performance with a higher ADG for crossbred black goats. Additionally, the mulberry leaf TMR diet promoted the deposition of IMF and IMP, which was associated with the changes in the expression of meat quality-related genes (*ADSL* and *H-FABP*). Consequently, a better quality of goat meat with an improved composition of fatty acids and amino acids was observed by feeding the mulberry leaf TMR diet to crossbred black goats. However, further investigation is needed to elucidate the underlying mechanisms. Notably, the results of this study suggested that the mulberry leaf TMR diet has the best effect on the meat quality when processed by pelleting in comparison with the control group, the freshly processed 40% mulberry leaf TMR diet and the mulberry leaf FTMR diet. In conclusion, feeding a 40% mulberry leaf pellet TMR diet to crossbred black goats is an effective and promising method to improve the meat quality of crossbred black goats.

## Figures and Tables

**Figure 1 foods-11-04032-f001:**
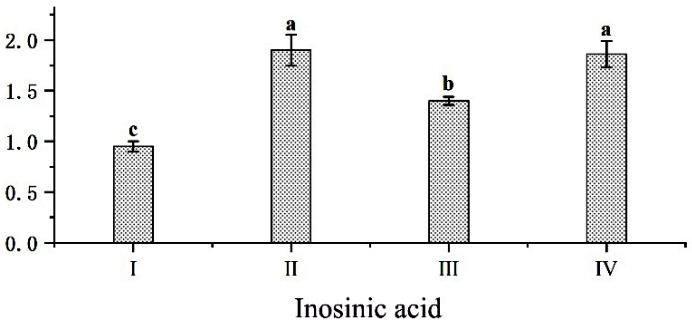
Effect of mulberry leaf TMR diet in different processing modes on black goat muscle inosinic acid. Different bars in the graph indicate different experimental groups. Different superscript lowercase letters in the same row (a, b, c) indicate significant differences (*p* < 0.01).

**Figure 2 foods-11-04032-f002:**
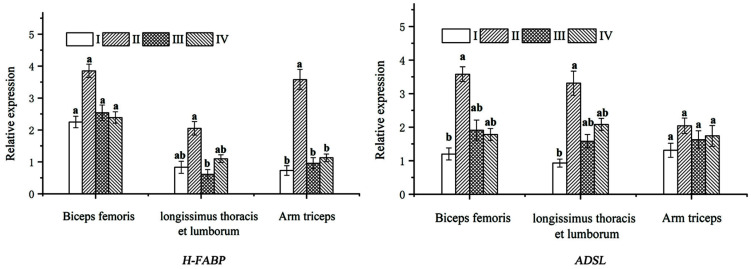
Effect of different processing methods of Morus alba TMR diet on the expression of *H-FABP* and *ADSL* genes in the muscle of black goats. Different bars in the graph indicate different experimental groups. Different superscript lowercase letters in the same row (a, b) indicate significant differences (*p* < 0.05).

**Table 1 foods-11-04032-t001:** Composition and nutrient levels of diets (%, air-dry basis).

Item	I	II	III	IV
Corn first grade	16.35	27.5	27.5	27.5
Bran	14.27	7.5	7.5	7.5
Soybean meal 43	11.4	4.3	4.3	4.3
yeast	1.98	2.1	2.1	2.1
salt	0.49	0.44	0.44	0.44
1% composite premix ^1^	0.99	0.7	0.7	0.7
baking soda	0.79	0.46	0.46	0.46
Peanut Vine	53.72	17	17	17
Fresh mulberry leaf	0	40	40	40
total	100	100	100	100
Chemical composition ^2^				
DM	80.83	91.54	77.00	59.17
NDF	38.65	41.48	41.94	36.57
CP	13.89	14.11	13.29	14.58
EE	2.05	2.17	2.17	2.81
Ash	8.36	10.25	10.11	10.66
Ca	0.63	0.74	0.77	0.86
P	0.39	0.36	0.4	0.35
ME, MJ/kg	11.04	11.04	11.04	11.04

^1^ Premix provides per kg of ration: Vitamin A 10,000 IU, Vitamin D 15,000 IU, Vitamin E 200 IU, Iron 1000–7500 mg, Zinc 5000–3750 mg, Copper 140–150 mg, Iodine 5–200 mg, Manganese 600–3000 mg, Cobalt 4–40 mg, Selenium 3.5–10 mg, calcium 4–20% and phosphorus 2–8%. ^2^ The rest are calculated values except DM, NDF, Ash EE and CP, which are measured values.

**Table 2 foods-11-04032-t002:** Fluorescence quantitative target gene primer-related information.

Gene	Primer Sequence (5’ → 3’)	Product Size, bp	Annealing Temperature, °C
*ADSL*	F: CCAGTTGGACCATTTGCT	131	55
	R: ATTCTGCCTTCACCTTCAT		
*H-FABP*	F: AGACCACGGCAGATGACA	113	56.5
	R: AACTATTTCCCGCACAAG		
*GAPDH*	F: CTGACCTGCCGCCTGGAGAAA	149	57
	R: GTAGAAGAGTGAGTGTCGCTT		

F: forward; R: reverse.

**Table 3 foods-11-04032-t003:** Effects of different TMR diets on the performance and economic benefits of black goats.

Items ^a^	Group				SEM	*p*-Value
	Ⅰ	Ⅱ	Ⅲ	Ⅳ		
DMI, g/d	1350.90 ^b^	1525.32 ^a^	1248.43 ^c^	1322.27 ^b^	0.01	<0.01
ADG, g/d	143.81 ^b^	175.24 ^a^	145.52 ^b^	155.24 ^ab^	7.61	0.023
FWR	9.65	8.73	9.03	8.66	0.06	0.084
Total weight gain, kg/one goat	10.79 ^b^	13.14 ^a^	10.91 ^b^	11.64 ^ab^	0.57	0.023
Feed cost, CNY/kg	1.92	2.45	1.50	1.61	—	—
Feed weight gain cost, CNY/kg	20.52 ^ab^	23.40 ^a^	18.62 ^b^	22.24 ^a^	1.08	0.031
Live black goat price, CNY/kg	56	56	56	56	—	—
Weight gain benefit, CNY/ one	388	429.51	407.35	397.22	30.85	0.085

Different lowercase letters in the same row (a, b, c) indicate significant/extremely significant (*p* < 0.05 or *p* < 0.01); see *p* values in each table for details. The same lowercase letters in the same row indicate non-significant differences (*p* > 0.05). The rest of the tables are expressed in the same way. SEM: standard error of the mean. kg/one goat: Kilograms per black goat. CNY/kg: CNY per kg. CNY/one: CNY per black goat.

**Table 4 foods-11-04032-t004:** Effects of TMR processing of mulberry leaf on apparent digestibility of black goats (%).

Item	Group				SEM	*p*-Value
	I	II	III	IV		
Dry matter (DM)	75.62 ^c^	82.24 ^a^	78.68 ^b^	76.28 ^c^	0.22	<0.01
Ether extract (EE)	80.26 ^b^	83.73 ^a^	82.78 ^ab^	81.98 ^ab^	0.72	0.046
Crude protein (CP)	76.92 ^b^	81.31 ^a^	78.79 ^ab^	78.04 ^ab^	0.68	<0.01
Neutral detergent fiber (NDF)	55.55 ^b^	58.73 ^a^	58.71 ^a^	57.12 ^ab^	0.73	0.04

Different lowercase letters in the same row (a, b, c) indicate significant/extremely significant (*p* < 0.05 or *p* < 0.01); see *p* values in each table for details. The same lowercase letters in the same row indicate non-significant differences (*p* > 0.05).

**Table 5 foods-11-04032-t005:** Effects of different processing methods of mulberry leaf TMR diet on muscle quality of black goats.

Item	Group				SEM	*p*-Value
	I	II	III	IV		
Longissimus thoracis et lumborum						
Lightness L*	36.67 ^a^	31.64 ^b^	36.14 ^a^	35.58 ^a^	0.078	<0.01
Redness a*	13.44	15.60	15.37	15.56	1.22	0.320
Yellowness b*	8.38 ^a^	5.53 ^b^	7.00 ^ab^	6.87 ^ab^	0.56	<0.01
pH_45 min_	7.33 ^a^	7.14 ^b^	7.21 ^ab^	7.17 ^b^	0.045	0.036
pH_24 h_	6.68	6.75	6.72	6.72	0.07	0.926
shear force, N	39.83 ^a^	37.95 ^b^	38.56 ^ab^	39.44 ^ab^	0.56	0.031
Water loss rate, %	83.05	84.07	83.57	83.51	0.72	0.802
cooking yield, %	46.21	50.44	47.84	48.85	1.77	0.438
Arm triceps						
Lightness L*	39.44 ^a^	37.95 ^b^	38.56 ^ab^	39.28 ^ab^	0.453	0.031
Redness a*	13.81	16.60	13.96	15.20	1.29	0.434
Yellowness b*	7.96 ^a^	4.07 ^b^	7.85 ^a^	6.05 ^ab^	0.781	<0.01
shear force, N	50.45 ^a^	35.18 ^b^	48.05 ^a^	44.37 ^ab^	3.39	0.017
Biceps femoris						
Lightness L*	42.85	37.81	40.36	38.76	2.58	0.562
Redness a*	13.33 ^b^	17.37 ^a^	14.06 ^ab^	15.62 ^ab^	1.23	0.031
Yellowness b*	9	6.11	8.68	6.53	1.16	0.219
shear force, N	51.72	43.51	49.49	44.96	3.35	0.287

Different lowercase letters in the same row (a, b) indicate significant/extremely significant (*p* < 0.05 or *p* < 0.01); see *p* values in each table for details. The same lowercase letters in the same row indicate non-significant differences (*p* > 0.05).

**Table 6 foods-11-04032-t006:** Effect of different processing modes of mulberry leaf TMR diet on the nutrient composition of black goat muscle (air-dry matter basis, %).

Item	Group				SEM	*p*-Value
	I	II	III	IV		
Moisture Content (MC)	27.50 ^b^	28.77 ^a^	28.15 ^ab^	28.56 ^a^	0.25	<0.01
Ether extract (EE)	6.80 ^b^	8.47 ^a^	7.66 ^ab^	8.00 ^a^	0.30	<0.01
Crude protein (CP)	20.38 ^b^	20.69 ^a^	20.46 ^b^	20.56 ^ab^	0.06	<0.01
Crude ash (Ash)	1.2	1.20	1.18	1.23	0.02	0.480

Different lowercase letters in the same row (a, b) indicate significant/extremely significant (*p* < 0.05 or *p* < 0.01); see *p* values in each table for details. The same lowercase letters in the same row indicate non-significant differences (*p* > 0.05).

**Table 7 foods-11-04032-t007:** Effect of different processing modes of mulberry leaf TMR diet on black goat muscle fatty acids.

Fatty Acid (g/100 g)	Group				SEM	*p*-Value
	I	II	III	IV		
C4:0	0.20 ^a^	0.13 ^b^	0.18 ^a^	0.19 ^a^	0.01	<0.01
C8:0	0.22 ^a^	0.18 ^b^	0.19 ^b^	0.18 ^b^	0.00	<0.01
C10:0	0.15	0.15	0.15	0.17	0.01	0.434
C11:0	0.07 ^a^	0.04 ^b^	0.04 ^b^	0.04 ^b^	0.00	<0.01
C12:0	0.11	0.12	0.08	0.14	0.02	0.267
C13:0	0.25	0.24	0.24	0.25	0.01	0.729
C14:0	1.82	2.03	1.86	1.92	0.09	0.380
C14:1	0.14 ^a^	0.10 ^b^	0.10 ^b^	0.08 ^b^	0.00	<0.01
C15:0	0.75 ^a^	0.54 ^bc^	0.43 ^c^	0.63 ^ab^	0.03	<0.01
C16:0	25.03 ^a^	19.18 ^b^	24.06 ^a^	19.54 ^b^	0.29	<0.01
C16:1	1.34 ^c^	3.20 ^a^	1.57 ^c^	2.67 ^b^	0.07	<0.01
C17:0	1.37	1.46	1.37	1.31	0.08	0.645
C17:1	0.92 ^bc^	1.41 ^a^	0.80 ^c^	1.10 ^b^	0.05	<0.01
C18:0	27.27 ^a^	20.32 ^b^	24.07 ^ab^	21.51 ^b^	1.10	<0.01
C18:1n9t	1.4	1.38	1.37	1.38	0.08	0.996
C18:1n9n	32.95 ^b^	50.71 ^a^	39.61 ^a^	39.43 ^a^	1.46	<0.01
C18:2n6t	0.13	0.16	0.15	0.15	0.01	0.193
C18:2n6	5.33 ^b^	7.13 ^a^	7.11 ^a^	7.76 ^a^	0.36	<0.01
C20:0	0.19	0.19	0.18	0.18	0.01	0.440
C18:3n6	0.73	0.07	0.71	0.61	0.01	0.437
C20:1	0.19	0.20	0.20	0.19	0.02	0.982
C18:3n3	0.65	0.77	0.73	0.66	0.08	0.619
C20:3n6	0.17 ^b^	0.33 ^a^	0.26 ^a^	0.33 ^a^	0.02	<0.01
C20:4n6	2.95 ^b^	4.34 ^a^	3.47 ^ab^	4.10 ^a^	0.29	0.037
Σ TFA	103.66 ^b^	114.37 ^a^	103.75 ^b^	108.47 ^ab^	3.20	0.018
Σ SFA	57.41 ^a^	44.58 ^c^	48.33 ^bc^	50.58 ^b^	0.88	<0.01
Σ USFA	46.25 ^b^	69.79 ^a^	55.41 ^a^	57.90 ^a^	1.74	<0.01
Σ MUFA	35.54 ^c^	55.61 ^a^	42.28 ^bc^	43.46 ^ab^	1.45	<0.01
Σ PUFA	9.07 ^b^	12.40 ^a^	11.43 ^ab^	12.66 ^a^	0.60	<0.01

Different lowercase letters in the same row (a, b, c) indicate significant/extremely significant (*p* < 0.05 or *p* < 0.01); see *p* values in each table for details. The same lowercase letters in the same row indicate non-significant differences (*p* > 0.05). SEM: standard error of mean. TFA: Total fatty acids. SFA: Total saturated fatty acids. USFA: Unsaturated fatty acids. MUFA: Monounsaturated fatty acids. PUFA: Polyunsaturated fatty acids. Σ TFA = Total sum of all fatty acids in the table. Σ SFA = C4:0 + C8:0 + C10:0 + C11:0 + C12:0 + C13:0 + C14:04 + C15:0 + C16:0 + C17:0 + C20:0 + C18:0. Σ USFA = C14:1 + C16:1 + C17:1 + C18:1n9t + C18:1n9n + C18:2n6t + C18:2n6 + C18:3n6 + C20:1 + C18:3n3 + C20:3n6 + C20:4n6. Σ MUFA = C14:1 + C16:1 + C17:1 + C18:1n9n + C20:1. Σ PUFA = C18:2n6t + C18:2n6 + C18:3n3 + C20:4n6.

**Table 8 foods-11-04032-t008:** The effect of different processing modes of mulberry leaf TMR diet on black goat muscle amino acids.

Amino Acids (g/100 g)	Group				SEM	*p*-Value
	I	II	III	IV		
Glycine (Gly)	3.39 ^b^	4.36 ^a^	3.94 ^ab^	4.00 ^a^	0.2	0.058
Alanine (Ala)	3.58 ^b^	4.43 ^a^	4.06 ^ab^	4.32 ^a^	0.17	0.029
Valine (Val)	2.78 ^b^	3.39 ^a^	3.12 ^a^	3.39 ^a^	0.12	0.021
Leucine (Leu)	4.82 ^b^	5.86 ^a^	5.36 ^ab^	5.87 ^a^	0.22	0.028
Isoleucine (Ile)	2.70 ^b^	3.17 ^a^	2.94 ^ab^	3.15 ^a^	0.09	0.014
Phenylalanine (Phe)	2.39 ^b^	2.94 ^a^	2.69 ^ab^	2.91 ^a^	0.11	0.025
Serine (Ser)	2.44 ^b^	2.99 ^a^	2.83 ^a^	2.97 ^a^	0.12	0.031
Tyrosine (Tyr)	2.00 ^b^	2.42 ^a^	2.18 ^ab^	2.47 ^a^	0.13	0.096
Aspartic Acid (Asp)	5.43 ^b^	6.59 ^a^	6.07 ^ab^	6.57 ^a^	0.25	0.032
Glutamic acid (Glu)	9.56 ^b^	11.56 ^a^	10.61 ^ab^	11.54 ^a^	0.49	0.059
Lysine (Lys)	5.21 ^b^	6.31 ^a^	5.78 ^ab^	6.30 ^a^	0.23	0.030
Methionine (Met)	1.28	1.45	1.32	1.69	0.23	0.595
Threonine (Thr)	2.73 ^b^	3.26 ^a^	3.06 ^ab^	3.32 ^a^	0.12	0.030
Cysteine (Cys)	0.23	0.26	0.29	0.29	0.32	0.565
Proline (Pro)	2.55 ^b^	3.22 ^a^	2.90 ^ab^	3.05 ^a^	0.14	0.045
Histidine (His)	1.95 ^b^	2.29 ^a^	2.03 ^ab^	2.34 ^a^	0.10	0.051
Arginine (Arg)	4.00 ^b^	4.94 ^a^	4.46 ^ab^	4.84 ^a^	0.20	0.038
Σ TAA	57.04 ^b^	68.37 ^a^	63.65 ^ab^	69.11 ^a^	2.55	0.035
Σ EAA	24.34 ^b^	29.68 ^a^	27.10 ^ab^	29.28 ^ab^	1.13	<0.01
Σ NEAA	32.70 ^b^	39.58 ^a^	36.55 ^ab^	39.92 ^a^	1.42	0.024
Σ DAA	27.97 ^b^	33.74 ^a^	31.33 ^ab^	34.31 ^a^	1.27	0.028
EAA/TAA, %	42.68	43.39	42.57	42.37	0.36	0.909
DAA/TAA, %	49.01 ^b^	49.33 ^b^	49.21 ^b^	49.69 ^a^	0.60	0.874

Different lowercase letters in the same row (a, b) indicate significant/extremely significant (*p* < 0.05 or *p* < 0.01); see *p* values in each table for details. The same lowercase letters in the same row indicate non-significant differences (*p* > 0.05). SEM: standard error of mean. TAA: Total amino acids. EAA: Total essential amino acids. NEAA: Total non-essential amino acids. DAA: Total of major fresh-tasting amino acids. Σ TAA = Total sum of all amino acids in the table. Σ EAA = Val + Leu + Ile + Phe + Ser + Lys + Met + Thr. Σ NEAA = Gly + Ala + Tyr + Asp + Glu + Cys + Pro + His + Arg. Σ DAA = Glu + Ala + Arg + Tyr + Asp + Gly.

## Data Availability

The data used to support the findings of this study can be made available by the corresponding author upon request.

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
