# Peer review of "Effect of Mulberry Leaf TMR on Growth Performance, Meat Quality and Expression of Meat Quality Master Genes (ADSL, H-FABP) in Crossbred Black Goats"

_foods, 2022, doi:10.3390/foods11244032_

Round 1
Reviewer 1 Report
The manuscript by Long et al deals with the application of mulberry leaves to the quality of goat meat. The manuscript is well written with easy and clear language which is easy to follow. The hypothesis should be presented with more clarity. The experimental design need to improve further.
My observations are as follows-
i. Title: The title does not represent the work carried out. I would like to expand the TMR in the heading and as meat quality indicates flavor also, so deleting flavor from the title. Further authors also evaluated the fattening effect/ digestibility, gene expression, feed intake, daily gains, etc. So, the present title should be edited to better represent the good work done by the authors.
ii. The scientific names in italics plz as L9- Morus alba
iii. In abstract: Authors should add more clarity on experimental design (as authors mentioned the group but without their detailing etc).
iv. There is a need to maintain uniformity of various terms as in some places authors used the term slaughter performance, daily feed gain, gene expression, so there is a need for improvement on this aspect.
v. L 98-99- authors used 28 animals whereas in abstract 44 animals, please correct.
vi. In the feed, there is a difference with other ingredients such as bran soy meal etc and the level of Mulberry leaves are 40%. So there is a need to add further justification that these effects could be due to other changes in feed ration in addition to Mulberry
vii. Also authors have described the effect of processing of mulberry, please detail if any processing is done
viii. L 156- correct the reference
ix. L 158- Authors used L lumborum or L. thoracic et lumborum
x. pH was taken on L. lumborum
xi. Table 10: too short may be presented in results and deleted
Author Response
Reviewer 1
- “Title: The title does not represent the work carried out. I would like to expand the TMR in the heading and as meat quality indicates flavor also, so deleting flavor from the title. Further authors also evaluated the fattening effect/ digestibility, gene expression, feed intake, daily gains, etc. So, the present title should be edited to better represent the good work done by the authors.”
Our response: Thanks for your valuable advice. We believe that the reviewer has considered the topic very comprehensively. After careful consideration, we have made modifications as the reviewer’s suggestion.
- "The scientific names in italics plz as L9-Morus alba"
Our response: Thanks for the correction. The manuscript has been double-checked and revised carefully.
- “In abstract: Authors should add more clarity on experimental design (as authors mentioned the group but without their detailing Etc)."
Our response: We greatly appreciate the reviewer’s comment. In the revised manuscript, we have made corresponding supplements and modifications to better describe our experimental design.
- “There is a need to maintain uniformity of various terms as in some places authors used the term slaughter performance, daily feed gain, gene expression, so there is a need for improvement on this aspect.”
Our response: Following the reviewer’s comment, the manuscript is revised to keep the consistency of the descriptions.
- “98-99- authors used 28 animals whereas in abstract 44 animals, please correct.",
Our response: we have proofread it carefully, and we have made corrections accordingly.
- “the feed, there is a difference with other ingredients such as bran soy meal etc and the level of Mulberry leaves are 40%. So there is a need to add further justification that these effects could be due to other changes in feed ration in addition to Mulberry"
Our response: Thank you very much for your suggestion. our previous study "Effects of TMR with different mulberry leaf Proportions on Fattening Effect of Black Goats Male Lambs", we found that TMR with 40% mulberry leaves has the best feeding effect [1]. Therefore, this study mainly focused on the effects of 40% mulberry leaf TMR on growth performance, slaughter performance and meat quality (fatty acids, amino acids, inosinic acid, etc.). Relevant studies have shown that the growth performance, nutrient digestibility and meat quality of animals can be improved after the addition of mulberry leaves or mulberry leaf powder. In addition, the aldehydes, alcohols, esters and ketones in mulberry leaves are the main characteristic compounds of meat flavor, and feeding mulberry leaves can improve the flavor of animal meat [2-3]. In addition to mulberry leaves, peanut vine, soybean meal and wheat bran also have some changes in our feed formula on the basis of fixed energy. We have reviewed relevant literature. In previous reports with similar variations to our formulations, it was found that there were no statistically significant changes in growth performance, muscle nutrients, muscle fatty acids, and amino acids in the formulations of peanut vine, soybean meal, and wheat bran in the feed compared to the control group. It indicates that the changes in feed soybean meal, wheat bran and peanut vine were not important factors affecting the results of the medium experiment, but the addition of mulberry leaves had a greater effect on the growth performance and meat products of the animals [4,5]. Following the reviewer’s comments, we have added relevant evidence in the discussion section in the revised manuscript.
- “Also authors have described the effect of processing of mulberry, please detail if any processing is done”
Our response: Thanks for your suggestion. More details of the processing methods have been added to the experimental design part of the revised manuscript.
- “L 156- correct the reference”
Our response: Thanks for pointing out the mistake, which is corrected in the revised manuscript.
- “L 158- Authors used L lumborum or L. thoracic et lumborum”
Our response: I apologized for the confusion. Muscle samples of “longissimus thoracis et lumborum” were used in this experiment. We have corrected the entail manuscript to make it consistent.
- “pH was taken on L. lumborum”
Our response: Thanks for your suggestion. Longissimus thoracis et lumborum was cut from the mid-section and the surface fascia was removed. A pH meter (pHS-3E. Shanghai Yidian Scientific Instruments Co. Ltd., China) was used to measure the pH value of longissimus thoracis et lumborum at 45 min and 24 h after slaughter.
- “Table 10: too short may be presented in results and deleted”
Our response: We greatly appreciate the reviewer’s comment. Inosinic acid in muscle, as a kind of umami substance, has become an important index to assess the quality of meat. The continuous accumulation of inosinic acid and other decomposition products can enhance meat taste and improve meat flavor [6]. Therefore, we believe that it is necessary to present this result. However, we have transformed the inosinic acid data (Table 10) to a bar chart (Figure 1 in the revised manuscript) to show the changes of inosinic acid in response to the treatments. Moreover, following the reviewer’s suggestion, Table 5 has been removed from the revised manuscript as the data did not reach the statistical meaning.
Reference
[1]. Chen, D.Q. Effect of TMR Diet on Fattening Effect of Black Goats [D]. Guizhou University. 2021.30-38. (in Chinese)
[2] HE H.L. Effcet of Mulberry Leaf Powder on the Production Performance, Slaughter Performance Meat Quality and Pork Flavor of pigs[D]. Gansu Agricultural University. 2018. (in Chinese)
[3] Zhao J, Wang M, Xie J, et al. Volatile flavor constituents in the pork broth of black-pig[J]. Food Chem, 2017, 226: 51-60.
[4] Zhang H, Zhang L, Xue X, et al. Effect of feeding a diet comprised of various corn silages inclusion with peanut vine or wheat straw on performance, digestion, serum parameters and meat nutrients in finishing beef cattle[J]. Animal Bioscience, 2021, 35: 29 - 38.
[5] Wang M Y, Han, H Q, S, Y, et al. Effect of the Replacement of Maize Silage and Soyabean Meal with Mulberry Silage in the Diet of Hu Lambs on Growth Performance, Serum Biochemical Indices, Slaughter Performance, and Meat Quality[J]. Animals.2022
[6] Iwamoto, Oka, Iwaki. Effects of the fattening period on the fatty acid composition of fat deposits and free amino acid and inosinic acid contents of the longissimus muscle in carcasses of Japanese Black steers[J]. Anim Sci J, 2009, 2009,80(4):411-417.
[7] Sun, H. Luo, Y. Zhao, F F.et al. The Effect of Replacing Wildrye Hay with Mulberry Leaves on the Growth Performance, Blood Metabolites, and Carcass Characteristics of Sheep[J]. Animals.2020
[8] Ma, J Y. Ma, H. Liu, S J, et al. Effect of Mulberry Leaf Powder of Varying Levels on Growth Performance, Immuno-Antioxidant Status, Meat Quality and Intestinal Health in Finishing Pigs[J]. Antioxidants.2022
Reviewer 2 Report
It is an interesting job. The methodology section requires improvements in the data analysis section. It is suggested to submit the data to normality tests. In addition, in the animal sacrifice section, it should be explained why the number is reduced from 7 to 5. The results section requires improvements. For example, figure 1 is unnecessary since it shows results without statistical significance. The tables must be reviewed and taken care of in their structure. It is suggested to reduce the number of tables (4 and 10 are unnecessary)
Author Response
“It is an interesting job. The methodology section requires improvements in the data analysis section. It is suggested to submit the data to normality tests. In addition, in the animal sacrifice section, it should be explained why the number is reduced from 7 to 5. The results section requires improvements. For example, figure 1 is unnecessary since it shows results without statistical significance. The tables must be reviewed and taken care of in their structure. It is suggested to reduce the number of tables (4 and 10 are unnecessary)”
Our response: Thanks you for carefully reviewing our manuscript and putting forward valuable advice.
The Shapiro-Wilk test was firstly applied to determine the normal distribution of the data, and all data are within the qualified range. We have added more descriptions in the revised manuscript. I apologize for the confusion about the number of animals used. In this study, each treatment group has 11 goats and we randomly selected 5 goats from each group for slaughter. The mistake has been corrected in the revised manuscript and thanks for pointing it out.
Figure 1 shows the expression of H-FABP gene in different muscle tissues of black goats. Please note that although H-FABP gene in biceps femoris muscle was not significantly different, it shows statistically different among groups in the longissimus thoracis et lumborum and the arm triceps. Therefore, we believe that Figure 1 is necessary to stay in the manuscript.
Following the reviewer’s suggestion, we have carefully reviewed and adjusted the structure of the tables in the revised manuscript.
The result of Table 4 demonstrated key findings of the effects of TMR processing methods on the apparent digestibility, where the difference is statistically significant so we think it is necessary to keep the data in Table 4. Table 5 shows the slaughter performance without significant findings. As suggested by the reviewer, the data in Table 5, along with its discussion, has been removed from the revised manuscript.
Inosinic acid in muscle, as a kind of umami substance, has become an important index to assess the quality of meat. The continuous accumulation of inosinic acid and other decomposition products can enhance meat taste and improve meat flavor [6]. Therefore, we believe that it is necessary to present this result. However, we have transformed the inosinic acid data (Table 10) to a bar chart (Figure 1 in the revised manuscript) to show the changes of inosinic acid in response to the treatments.
Reference
[1]. Chen, D.Q. Effect of TMR Diet on Fattening Effect of Black Goats [D]. Guizhou University. 2021.30-38. (in Chinese)
[2] HE H.L. Effcet of Mulberry Leaf Powder on the Production Performance, Slaughter Performance Meat Quality and Pork Flavor of pigs[D]. Gansu Agricultural University. 2018. (in Chinese)
[3] Zhao J, Wang M, Xie J, et al. Volatile flavor constituents in the pork broth of black-pig[J]. Food Chem, 2017, 226: 51-60.
[4] Zhang H, Zhang L, Xue X, et al. Effect of feeding a diet comprised of various corn silages inclusion with peanut vine or wheat straw on performance, digestion, serum parameters and meat nutrients in finishing beef cattle[J]. Animal Bioscience, 2021, 35: 29 - 38.
[5] Wang M Y, Han, H Q, S, Y, et al. Effect of the Replacement of Maize Silage and Soyabean Meal with Mulberry Silage in the Diet of Hu Lambs on Growth Performance, Serum Biochemical Indices, Slaughter Performance, and Meat Quality[J]. Animals.2022
[6] Iwamoto, Oka, Iwaki. Effects of the fattening period on the fatty acid composition of fat deposits and free amino acid and inosinic acid contents of the longissimus muscle in carcasses of Japanese Black steers[J]. Anim Sci J, 2009, 2009,80(4):411-417.
[7] Sun, H. Luo, Y. Zhao, F F.et al. The Effect of Replacing Wildrye Hay with Mulberry Leaves on the Growth Performance, Blood Metabolites, and Carcass Characteristics of Sheep[J]. Animals.2020
[8] Ma, J Y. Ma, H. Liu, S J, et al. Effect of Mulberry Leaf Powder of Varying Levels on Growth Performance, Immuno-Antioxidant Status, Meat Quality and Intestinal Health in Finishing Pigs[J]. Antioxidants.2022
Reviewer 3 Report
1. Title
- Experimental black goat is pure breed. Please chage "black goat meat" into "crossbred goat meat".
2. Abstract
- Please explain about all groups more in details.
- Keywords: Black goat -> Goat meat
3. Results and conclusions
- Authors described that Mulberry leaf TMR diet promoted the doposition of intramuscular fat and increased heathy polyunsasutrated. In general, high marbled meat contains high concentration of monounsaturated fatty acids, not polyunsaturated fatty acids. But your reaserch results are opposite to general theory. How will you explain this phenomenon?
Author Response
Reviewer 3
- About "Experimental black goat is pure breed. Please chage "black goat meat" into "crossbred goat meat"
Our response: Thank you for the suggestion. Modifications have been made as the reviewer’s suggestion.
- For the question "Please explain about all groups more in details."
Our response: Thanks for your comment. More details have been added in the revised manuscript as suggested by the reviewer.
- “Keywords: Black goat -> Goat meat”
Our response: Following the reviewer’s comment, we have made the modification in the revised manuscript.
- “Authors described that Mulberry leaf TMR diet promoted the doposition of intramuscular fat and increased heathy polyunsasutrated. In general, high marbled meat contains high concentration of monounsaturated fatty acids, not polyunsaturated fatty acids. But your reaserch results are opposite to general theory. How will you explain this phenomenon?”
Our response: We greatly appreciate the reviewer’s comments. Through our review of many sources of literature, it is true that high marbled meat contains high concentration of monounsaturated fatty acids, not polyunsaturated fatty acids. It is clear from the data represented in this study that feeding of 40% mulberry leaf TMR caused some degree of change in muscle fatty acids in black goats and promoted intermuscular fat deposition. In response to the treatment, monounsaturated fatty acid and polyunsaturated fatty acids did show an increasing trend in the meats. However, the content of monounsaturated fatty acids was still higher than that of the Polyunsaturated fatty acids, which is consistent with previous studies [7,8] and follows the general theory.
Round 2
Reviewer 1 Report
The present manuscript is improved a lot. I am thankful to the authors for their thorough editing and revision. To improve the quality and widen readership, I have the following suggestions-
i. Title: Now it’s clearer and better. Author may use ‘Effect of Mulberry Leaf TMR on Growth Performance, Meat Quality and Expression of Meat Quality Master Genes (ADSL, H-4 FABP) in Crossbred Black Goats’
ii. Abstract: Authors have improved the abstract, but I feel it is too wordy (approx. 530 word) and may not fit the standard format of the Foods. Please concise it.
iii. L58: inosinic acid/ inosine monophosphate (IMP)
iv. L181-182: Shear force and water loss rate; Do authors referring water rate loss referred to Drip loss (loss of exudates upon refrigeration) or shrinkage (loss of moisture)? Further please check Garba et al (31for these parameters.
v. L 183: plz check the reference, may be Aa et al. (32)
vi. L456: cooked meat rate may be replaced with cooking yield
vii. Results, discussion and conclusion: Appropriate
Author Response
- Title: Now it’s clearer and better. Author may use ‘Effect of Mulberry Leaf TMR on Growth Performance, Meat Quality and Expression of Meat Quality Master Genes (ADSL, H-FABP) in Crossbred Black Goats’
sincerely appreicate the reviewer's recognition. Following the reviewer's suggestion, the revised the title to “Effect Mulberry Leaf TMR on Growth Performance, Meat Quality and Expression of Meat Quality Master Genes (ADSL, H-FABP) in Crossbred Black Goats” which is more concise.
- “Abstract: Authors have improved the abstract, but I feel it is too wordy (approx. 530 word) and may not fit the standard format of the Foods. Please concise it.”
Our response: We thank the reviewer’s comments and have refined the abstract to make it concise and informative.
- L58: inosinic acid/ inosine monophosphate (IMP)
Our response: Thank you for the suggestion. Inosine monophosphate (IMP) was analyzed as inosinic acid in this study. The reviewer is correct that we’d better clarify this.
- L181-182: Shear force and water loss rate; Do authors referring water rate loss referred to Drip loss (loss of exudates upon refrigeration) or shrinkage (loss of moisture)? Further please check Garba et al (31for these parameters.
Our response: We greatly appreciate the reviewer’s comments. The shear force in this study was detected according to the method of Garba et al [31]. We have carefully checked and found that a reference for water loss rate detection was missing in the original manuscript. In this revision, we have added the missed reference and carefully checked the references in the whole manuscript again. For the water loss rate, we used the pressure shrinkage method [33] instead of Drip loss (loss of exudates upon refrigeration). Thanks for pointing it out.
- L 183: plz check the reference, may be Aa et al. (32)
Our response: Thanks for pointing out this typo, which is now corrected in the revised manuscript.
- L456: cooked meat rate may be replaced with cooking yield
Our response: Thank you for the valuable advice. “Cooking yield” is indeed a better description. Modifications have been made as the reviewer’s suggestion.
